# The Influence of Circadian Rhythm on the Activity of Oxidative Stress Enzymes

**DOI:** 10.3390/ijms232214275

**Published:** 2022-11-17

**Authors:** Marta Budkowska, Elżbieta Cecerska-Heryć, Zuzanna Marcinowska, Aldona Siennicka, Barbara Dołęgowska

**Affiliations:** 1Department of Medical Analytics, Pomeranian Medical University, 70-111 Szczecin, Poland; 2Department of Laboratory Medicine, Pomeranian Medical University, 70-204 Szczecin, Poland; 3Department of Biochemistry and Medical Chemistry, Pomeranian Medical University, 70-111 Szczecin, Poland

**Keywords:** circadian rhythm, antioxidant enzymes, oxidative stress

## Abstract

The circadian system synchronizes daily with the day–night cycle of our environment. Disruption of this rhythm impacts the emergence and development of many diseases caused, for example, by the overproduction of free radicals, leading to oxidative damage of cellular components. The goal of this study was to determine the activity of superoxide dismutase (SOD), glutathione peroxidase (GPx), catalase (CAT), glutathione transferase (GST), glutathione reductase (R-GSSG), and the concentration of glutathione (GSH) in the circadian rhythm. The study group comprised 66 healthy volunteers (20–50 years; 33 women; 33 men). The blood was collected at 2, 8 a.m., and 2, 8 p.m. All samples marked the serum melatonin concentration to confirm the correct sleeping rhythm and wakefulness throughout the day. The activity of study enzymes and the concentration of GSH were measured by the spectrophotometric method. Confirmed the existence of circadian regulation of oxidative stress enzymes except for GST activity. The peak of activity of study enzymes and GSH concentration was observed at 2 a.m. The increased activity of enzymes and the increase in GSH concentration observed at night indicate that during sleep, processes allowing to maintain of the redox balance are intensified, thus limiting the formation of oxidative stress.

## 1. Introduction

All organisms on Earth developed a mechanism of cyclical and repeating daily changes depending on the rhythm of the day and night, which influence the regulation of the activity of cells, tissues, and organs. The integrity of the circadian clocks and their synchronization with cellular processes is crucial for human health. It harmonizes our body’s physiological, mental, and behavioral functions, particularly the rhythm of sleep and wakefulness [1]. Disruption of such a rhythm, especially insufficiently long sleep, is a risk factor for premature aging, obesity, or depression. It also affects the emergence and development of many other diseases, including neurodegenerative diseases, cancer, or cardiovascular system dysfunction [2,3,4]. Studies on birds and mammals have shown that melatonin regulates the circadian clock both in vitro and in vivo [5,6]. It has also been shown that melatonin influences circadian rhythms in humans and other mammals by acting directly on receptors in the suprachiasmatic nucleus (SCN) [7]. Studies have also shown that melatonin is one of the most potent antioxidants [8,9]. In the brain, melatonin is oxidized to N1-acetyl-N2-formyl-5-methoxycynuramine (AFMK). This compound is also formed after melatonin combines with the reactive oxygen species (ROS) [10,11]. AFMK has an antioxidant capacity in vitro and is a compound that participates in the process in which melatonin and its metabolites successively remove free radicals and ROS and stimulate various antioxidant enzymes such as glutathione peroxidase (GPx), glutathione reductase (R-GSSG), superoxide dismutase (SOD), and catalase (CAT) [12,13].

Over millions of years of evolution, organisms have developed various protective systems to control ROS overactivity and the accompanying damage at the tissue level or in individual cells (lipid peroxidation, DNA, and protein damage), which contributes to the formation of oxidative stress [14,15]. The rise in levels of ROS can be triggered by many factors, such as lack of exercise plays, processed food, exposure to a wide range of chemicals, and also alcohol and toxins. Disturbance in the balance between the antioxidant protection system and the production of ROS can influence the formation of many diseases such as diabetes, cardiovascular diseases including myocardial infarction, stroke, neurodegenerative diseases, and cancer [16,17]. These preventive defense systems, primarily antioxidant defense, involve several strategies, both enzymatic and non-enzymatic [8,17,18,19,20,21,22]. The main non-enzymatic antioxidants include glutathione (GSH) [21,22], vitamins, especially vitamin C [23,24] and E [24,25,26], selenium [27], carotenoids [28,29], thioredoxins (Trx) [30,31], lipoic acid (ALA) [32] and flavonoids [33]. In turn, the presence and activity of cellular antioxidant enzymes are the basic defense mechanism against ROS, the most important of which are SOD, CAT, GPx, R-GSSG, and glutathione transferase (GST) [15]. Their primary function is to prevent reactive tissue damage caused by the oxygen radical. Their role is to prevent the formation of ROS and, when they are generated, neutralize them to inactive compounds [18]. Microarray studies suggest daily changes in the activity of some antioxidant enzymes, such as SOD, CAT, or GST [34]. It is postulated that such modifications may protect the body against the formation of excessive amounts of ROS and the resulting damage to tissues or cells [35].

This study aimed to determine whether the activity of selected oxidative stress enzymes (CAT, SOD-1, GPx, R-GSSG, GST) and the concentration of GSH in healthy volunteers is regulated in the circadian rhythm and to correlate these activities with the concentration of melatonin at individual time points of the collection. Such a simultaneous determination of the daily melatonin profile with the registration of the activity of antioxidant enzymes can be an excellent diagnostic element for detecting, among others, sleep disorders related to the circadian rhythm. An additional advantage of our study is the determination of antioxidant enzymes and the concentration of GSH in humans, as the studies have been based mainly on animal analysis.

## 2. Results

### 2.1. Morphological, Biochemical, and Blood Minerals Parameters

When analyzing the obtained results of morphological, biochemical, and blood minerals parameters, it was noticed that they differed from the established reference values. Primary statistical differences in the results between the groups were divided by gender. The differences concerned the following parameters: RBC (*p* < 0.0001), Hb (*p* < 0.0001), TG (*p* = 0.0207), CRE (*p* = 0.0012), UA (*p* = 0.0005). The other parameters did not show any significant statistical differences. Table 1 presents the arithmetic mean and the standard deviation of the test results for women and men, as well as the differences between them expressed by the level of significance (*p*).

### 2.2. Analysis of the Influence of the Circadian Rhythm on the Concentration of Melatonin in the Blood Serum

In the conducted research, it was noticed that the average concentration of melatonin is the highest around 2 a.m. regardless of gender. It was 99.5 pg/mL in women and 97.1 pg/mL in men. The lowest concentrations of this hormone were recorded at 2 p.m. The concentration in women was 8.7 pg/mL, and in men, 8.0 pg/mL (Table 2). The MANOVA analysis did not confirm statistically significant differences depending on gender (Figure 1). The analysis performed using the Friedman ANOVA test found that the difference in melatonin concentrations for the population presented by the given samples between at least two hours in which the tests were performed is statistically significant. However, the analysis of the Kendall concordance coefficient confirms the consistency in the measurements in the given hours in both women and men. On this basis, we can therefore confirm the occurrence of the circadian rhythm for the melatonin we are testing. Statistical analysis was also extended to the Wilcoxon order of pair test, which confirmed the presence of statistically significant differences between all time points. Each pair of the collection hours compared with each other showed a statistically significant correlation in the group of women and men at the *p* < 0.0001.

### 2.3. Analysis of the Influence of Circadian Rhythm on the Activity of Superoxide Dismutase (SOD-1)

In the conducted research, it was observed that the average activity of SOD-1 in women and men is the highest at 2 a.m. (in women—546 U/g Hb, in men—542 U/g Hb). The average lowest activity of this enzyme was recorded at 2 p.m. (in women—444 U/g Hb, in men—480 U/g Hb) (Table 3). The MANOVA analysis did not confirm statistically significant differences between genders (Figure 2). The study using the Friedman ANOVA test found that the difference in SOD-1 activity for the population presented by the given samples, between at least two analyzed hours, is statistically significant. On the other hand, Kendall’s coefficient of concordance analysis showed agreement in the measurements at given hours in both women and men. Therefore, based on the performed statistical tests, the daily changes in SOD-1 activity were confirmed. The statistical analysis was also extended to Wilcoxon’s pairwise order test, which confirmed the presence of statistically significant differences between all time points. Each pair of sampled hours of the collection showed a statistically significant correlation in the group at *p* < 0.0001. Spearman’s rank coefficient measured the strength of the correlation between melatonin and SOD-1, but no correlation was observed in this case.

### 2.4. Analysis of the Influence of the Circadian Rhythm on the Activity of Catalase (CAT)

The research observed that the average CAT activity in women and men is the highest around 2 a.m. (185 U/g Hb in women, 181 U/g Hb in men). The mean lowest activity of this enzyme was recorded 18 h later, i.e., at 8 p.m. (145 U/g Hb in women, 150 U/g Hb in men) (Table 4). The MANOVA analysis did not confirm statistically significant differences between genders (Figure 3). The study using the Friedman ANOVA test found that the difference in CAT activity for the population presented by the given samples between at least two analyzed hours is statistically significant. On the other hand, Kendall’s coefficient of concordance analysis showed agreement in the measurements at given hours in both women and men. Therefore, based on the performed statistical tests, daily changes in CAT activity were confirmed. The statistical analysis was also extended to Wilcoxon’s pairwise order test, which confirmed the presence of statistically significant differences between all time points. Each pair of sampled hours of the collection showed a statistically significant correlation in both the group of women and men, at the level of *p* < 0.0001. The Spearman rank coefficient measured the strength of the correlation between melatonin and CAT. Still, no correlation was observed in this case, except for one, between melatonin concentration in women at 8 a.m. and the CAT activity in men at 8 p.m. (Rs = 0.3617, *p* = 0.0386).

### 2.5. Analysis of the Influence of Circadian Rhythm on the Activity of Glutathione Peroxidase (GPx)

In the research, we observed that the average activity of GPx in women and men is the highest around 2 a.m. (8.2 U/g Hb in women, 8.8 U/g Hb in men). We recorded the average lowest activity of this enzyme was recorded at 2 p.m. (7.0 U/g Hb in women, 7.5 U/g Hb in men) (Table 5). The MANOVA analysis did not confirm statistically significant differences depending on gender (Figure 4). A study using the Friedman ANOVA test found that the difference in GPx activity for the population presented by the given samples, between at least two analyzed hours, is statistically significant. On the other hand, Kendall’s coefficient of concordance analysis showed agreement in the measurements at given hours in both women and men. Therefore, based on the performed statistical tests, we confirmed daily changes in GPx activity. We also extended the analysis to Wilcoxon’s pairwise order test, which confirmed the presence of statistically significant differences between all time points. Each pair of sampled hours of the collection showed a statistically significant correlation, both in the group of women and men, at the level of *p* < 0.0001. Spearman’s rank factor measured the strength of the correlation between melatonin and PGx, but no correlation was observed in this case.

### 2.6. Analysis of the Influence of Circadian Rhythm on the Activity of Glutathione Reductase (R-GSSG)

In the conducted research, it was observed that the average activity of R-GSSG in both women and men is the highest around 2 a.m. (1.4 U/g Hb in both women and men). The mean lowest activity of this enzyme was recorded 18 h later, i.e., at 8 p.m. (1.0 U/g Hb in both women and men) (Table 6). The MANOVA analysis did not confirm statistically significant differences depending on gender (Figure 5). A study using the Friedman ANOVA test found that the difference in R-GSSG activity for the population presented by the given samples between at least two analyzed hours is statistically significant. On the other hand, Kendall’s coefficient of concordance analysis showed agreement in the measurements at given hours in both women and men. Therefore, based on the performed statistical tests, the daily changes in the activity of R-GSSG were confirmed. The statistical analysis was also extended to Wilcoxon’s pairwise order test, which confirmed the presence of statistically significant differences between all analyzed time points. Each pair of sampled hours showed a statistically significant correlation, both in the women and men groups, at *p* < 0.0001. The Spearman rank coefficient measured the strength of the correlation between melatonin and R-GSSG. Still, no correlation, in this case, was observed except for one between melatonin concentration and R-GSSG activity in women at 8 a.m. (Rs = 0.3468, *p* = 0.0480).

### 2.7. Analysis of the Influence of Circadian Rhythm on the Activity of Glutathione Transferase (GST)

In the conducted research, it was observed that the average activity of GST in women and men is the highest at 2 a.m. (in women—0.44 U/g Hb, in men—0.40 U/g Hb). The mean lowest activity of this enzyme was recorded at 2 p.m. (in women—0.34 U/g Hb, in men—0.34 U/g Hb) (Table 7). The MANOVA analysis did not confirm statistically significant differences depending on gender (Figure 6). In the case of both genders, the analysis of the ANOVA Friedman test and the Kendall coefficient of concordance showed no statistically significant differences for the GST activity in the population presented by the given samples between the analyzed sampling hours. Thus, based on the performer statistical tests, the daily change in GST activity was not confirmed. The statistical analysis was also extended to the Wilcoxon pairwise test, which did not confirm the presence of statistically significant differences between all analyzed time points. Spearman’s rank factor measured the strength of the correlation between melatonin and GST. In this case, two correlations were observed—between the activity of GST in men and the concentration of melatonin in women at 8 a.m. (Rs = 0.4466, *p* = 0.009) and between the concentration of melatonin in men at 8 p.m. and the activity of GST in men at 8 a.m. (Rs = 0.4240, *p* = 0.0139).

### 2.8. Analysis of the Influence of the Rhythm on the Concentration of Glutathione (GSH)

In the conducted research, it was observed that the average concentration of GSH in both women and men is the highest around 2 a.m. (9.0 mM/L in women, 9.5 mM/L in men). The mean lowest GSH concentration was recorded 18 h later, i.e., at 8 p.m. (7.0 mM/L in women, 7.5 mM/L in men) (Table 8). The MANOVA analysis did not confirm statistically significant differences depending on gender (Figure 7). The study using the Friedman ANOVA test found that the difference in GSH concentration for the population presented by the given samples, between at least two analyzed hours, is statistically significant. On the other hand, Kendall’s coefficient of concordance analysis showed agreement in the measurements at given hours in both women and men. Therefore, based on the performed statistical tests, daily changes in GSH concentration were confirmed. The statistical analysis was also extended to Wilcoxon’s pairwise order test, which confirmed the presence of statistically significant differences between all time points. Each pair of sampled hours of the collection showed a statistically significant correlation, both in the group of women and men, at the level of *p* < 0.0001. Spearman’s rank factor measured the strength of the correlation between melatonin and GSH. In the case of GSH, two correlations were observed—between the concentration of GSH in men at 8 a.m. and the concentration of melatonin in women at 2 a.m. (Rs = 0.3578, *p* = 0.040) and between the concentration of GSH in men and the concentration of melatonin in women at 8 p.m. (Rs = −0.3662, *p* = 0.0361).

## 3. Discussion

More and more scientific studies show that the interaction of circadian rhythms with cellular processes promotes the biological regeneration of organisms and thus reduces their aging [2]. In recent years, it has also been proven that the aging process increases oxidative stress in the body caused by the physiological and non-physiological production of free radicals [35]. The basis of the defense system of each organism to fight oxidative stress is antioxidant enzymes, of which CAT, SOD, GPx, R-GSSG, and GST have the greatest influence [18]. The oxidative state of the body is also influenced by many other antioxidant factors, including GSH [12]. Our study aimed to determine whether, in healthy people, there are cyclical changes in the activity of the above-mentioned oxidative stress enzymes and changes in GSH concentration during the day.

### 3.1. Morphological, Biochemical, and Blood Minerals Parameters

The basic blood tests’ performance included complete blood counts, basic biochemical parameters, and tests to assess the calcium-phosphate and magnesium metabolism. The obtained results were analyzed statistically, taking into account the gender of the volunteers. In most cases, this analysis did not show statistically significant differences. However, for several parameters (RBC, Hct, Hb, CRE, UA, and TG), such gender differences were noted. The statistically significant differences depending on gender that we have noticed in the case of red cell morphological parameters (RBC, Hb, Hct) are most likely due to the different physiology of women and men. Red cell parameters can be influenced by muscle mass, adipose tissue content, water consumed, and circulating blood volume [36]. Studies have also shown that the main factor determining the differentiation between women and men, in this case, is the level of erythropoietin (EPO)—an erythropoiesis-stimulating hormone produced in the kidneys [37]. It is probably the only erythropoietic inducer in hypoxia situations (e.g., hemolysis, blood loss, poisoning, and a long stay at high altitudes) [37,38]. Many studies show that the action of testosterone stimulates its synthesis. Therefore, higher values of red cell parameters occur in men [39,40]. Apart from the level of erythropoietin increases in response to the state of hypoxia, its secretion during the day is normally characterized by a rhythmic pattern—the highest concentration is reached around 8 p.m., and the lowest is observed around 8 a.m. [41]. In our studies, we also showed gender-related statistically significant differences in the case of TG. A statistically significant difference between the genders may be the result from a varied diet. Higher TG concentrations are observed in people who eat a diet rich in carbohydrates and fats, which is more common in men [38,42]. The next parameters in which we observed gender-related differences that are statistically significant are CRE and UA. In the case of CRE, the differences between men and women are due to many factors. Among them, we can mention the diet (men consume larger amounts of meat products, favoring an increase in blood creatinine concentration by up to 30% and total muscle mass). On the other hand, the concentration of UA depends on age and gender, but also on renal function. The amount of UA produced results from the balance between the production and the intensity of the degradation of purines, which may arise due to increased physical effort (this usually occurs with greater intensity in men) [36,38,42].

### 3.2. The Circadian Rhythm of Melatonin: Influence of Melatonin Concentration on the Activity of Tested Oxidative Stress Enzymes 9SOD-1, CAT, GPx, R-GSSG, CST) and GSH Concentration

The secretion of melatonin takes place in the daily cycle. Its endogenous secretion is regulated by the light/dark cycle [43,44]. The highest levels of this hormone occur at night, between 10 p.m. and 6 a.m. Light stimuli cause a decrease in melatonin secretion and thus lower its concentration in the blood. Many scientists believe that melatonin is the elixir of youth because it protects the body against aging and has also been popularized as a cure for jet lag [45]. Our research determined melatonin concentration at four time points covering one daily cycle. The volunteers were provided with adequate sleep and wake conditions to obtain reliable results. It has been observed that the highest melatonin peak in both women and men is at 2 a.m. The lowest melatonin concentration was observed in the afternoon, around 2 p.m. The statistical analysis confirmed statistically significant differences between the individual sampling times, which was consistent with the previously described observations regarding the daily regulation of the secretion of this hormone in the blood [5,6]. The finding of such differences confirmed that our volunteers maintained normal sleep and wake rhythms throughout the entire period of the ongoing study. Despite the increasing number of reports that melatonin inhibits oxidative stress and stimulates enzymes that are antioxidants [46], in our research, we did not notice any significant correlation between melatonin and the oxidative stress enzymes studied by us in the context of their daily regulation. We indeed observed single correlations between the concentration of melatonin and the activity of CAT, R-GSSG, and GST or the concentration of GSH. Due to the inconsistency of their occurrence, we consider them rather random. For example, they may be a correlation resulting only from individual differences.

### 3.3. The Circadian Rhythm of SOD-1

SOD enzyme is an important factor involved in the defense of cells against oxidative stress caused by free radicals. This enzyme is considered to be the first line of defense against ROS because its main function in all aerobic organisms is to neutralize O_2^−^_ produced in the cytosol, mitochondria, and the endoplasmic reticulum of cells [47,48]. SOD, however, can also act pro-oxidative as O_2^−^_ dissociation causes the production of H_2_O_2_, which is toxic to cells. To remove dangerous H_2_O_2_, the presence of other antioxidant systems is necessary, such as the enzymes CAT (decomposition of H_2_O_2_ into H_2_O and O_2_) and GPx (reduction of H_2_O_2_ to H_2_O or appropriate alcohols using reduced GSH as an electron donor) [49,50,51,52]. The isoform of SOD1–copper–zinc dismutase is present in the cytosol and the cell nucleus, which distinguishes them from SOD-2 and SOD-3 [48]. Thus, the distinct isoform for erythrocytes we used in our research was the SOD-1 isoform.

The results of the own research showed that both in the case of women and men, the average SOD-1 activity is the highest at 2 a.m. Comparing our results with the first studies of this type that were carried out on rats, a certain similarity can be noticed [51]. Depending on the tests (whether they were carried out on the cerebral cortex, lungs, or intestines), the peak of SOD activity was at night [51,52]. The cerebral cortex was most active between 8 p.m., and 4 a.m., while the rat intestine was active between 9 p.m. and midnight. So, it seems that the daily SOD activity in rats and humans is similar, but the situation still needs to be viewed with a dose of distrust. Rats are only model organisms and do not always reflect all the processes in the human body. Therefore, such studies were also carried out in humans, but their results differ from those in our research. In studies conducted on a group of 60 clinically healthy volunteers and 30 patients with cirrhosis of the liver aged 25–45 years, the highest activity of SOD-1 was shown at 6 a.m., while the lowest activity was at 6 p.m. In both cases, the peaks coincided, but the enzyme values were lower in patients with cirrhosis compared to the healthy group of volunteers. Scientists showed a clear diurnal difference in SOD activity in humans. Still, their research also showed a general decline in defense mechanisms in patients with cirrhosis, indicating the participation of free radicals in the etiopathogenesis of the disease [53]. Comparing these results, we can observe that both the highest and the lowest activity of the enzyme in our research occurred 4 h earlier. The study of these scientists and their research was carried out in erythrocytes, so this excludes differences in the biological material used. On the other hand, the differences between the test results may be adopted different collection times (6 a.m., 12 a.m., 6 p.m., and 12 p.m.), which does not exclude that if we had taken the measurements at the same hours, we would have received very similar results. In addition, these scientists did not include the method they used to determine SOD-1 in the publication. This allows us to assume that the technique differed from ours. Our research and earlier human studies [53] confirmed the diurnal regulation of SOD-1 activity. It is worth noting that the peak of activity occurs at night. Scientists’ reports confirm that maintaining proper body function and adequate sleeping time is needed to remove the ROS accumulated throughout the day, despite more and more information appearing.

### 3.4. The Circadian Rhythm of CAT

Catalase is needed to maintain the integral antioxidant part, transforming the obtained, among others, during SOD H_2_O_2_ to H_2_O and O_2_. CAT is mainly located in peroxisomes. However, it has been proven that it is involved in protecting erythrocytes against H_2_O_2_ [49].

The results of our research showed that both in the case of women and men, the average CAT activity is the highest at 2 a.m. In the results obtained by a group of other scientists who first conducted such studies in 2005 on clinically healthy volunteers and patients with cirrhosis of the liver aged 25–45 years, it can be noticed that the highest activity of CAT occurs at 6 a.m. [53]. Comparing these results with ours, we can see that in our research, the peak of activity occurred earlier at 2 a.m. A similar situation occurred when it came to the lowest activity. According to the researchers, the most insufficient CAT activity was observed at midnight, and in the case of our research, 4 h earlier, i.e., at 8 p.m. The study was carried out in erythrocytes, but these scientists used a different method of determination, which was not included in the publication, then in their research. In addition, these scientists adopted other download times (6 p.m., 12 a.m., 6 p.m., and 12 p.m.). There is a high probability that if we had performed the downloads at the same hours, we would have obtained similar results because the upward and downward trends were identical depending on the time of day/night. As in the case of SOD-1 activity tests, CAT activity peaked at night. Scientists’ reports confirm that maintaining proper body function and adequate sleeping time is needed to remove the ROS accumulated throughout the day. The results of our research and that of other scientists [53] indicate the presence of diurnal regulation in CAT activity.

### 3.5. The Circadian Rhythm of GPx

The antioxidant enzyme task is to reduce H_2_O_2_ to H_2_O or appropriate alcohols with the help of reduced glutathione and selenium is GPx [54].

The results of our own research showed that both in the case of women and men, the average GPx activity is the highest at 2 a.m. The results obtained during the studies carried out in Swiss Webster mice differed from the results obtained in our study [55]. The maximum activity of this enzyme in Swiss Webster mice occurred at 2 p.m. and 6 a.m, so the time difference was up even to 12 h. The lowest activity was recorded at 10 a.m. and 8 p.m. and the results of our research showed that it was at 2 p.m. Comparing our results with those of the researchers, it can be seen that the lowest activity occurred 4 h earlier or 6 h later than the activity found by scientists in mice of this strain. Interestingly, the results obtained 2 years later confirmed the circadian rhythmicity in the rat cerebral cortex with the peak of activity in the evening and at night (8 p.m. and 4 a.m.) [51], which was much more similar to the results obtained by our group. This time, subsequent research results on clinically healthy volunteers and patients with cirrhosis of the liver aged 25–45 years also differed from our results [53]. In these studies, the same trend of increases and decreases in the peaks of GPx activity was observed in one group and the other group. The highest value was at 6 p.m. and the lowest at midnight, which differed by 8 h compared to our research. Nadir was due at midnight; compared to our results, the difference was 14 h. The differences in the results of studies carried out on rats can be explained by the fact that the rat is only a model organism, which does not always reflect the processes in the human body. On the other hand, we can exclude differences in the material studied in human studies because, as in our research, they were carried out in erythrocytes. These scientists did not include information on the method of measuring enzyme activity in the publication, which allows us to assume that it differed from ours. We noticed that the peak of activity occurs at night, i.e., during sleep. Sleep is essential for the proper functioning of the body because it is during sleep that repair processes occur, and these are the processes aimed at removing ROS accumulated throughout the day. Despite the differences in the results of the studies, both other scientists [55] and us, after conducting statistical analysis, showed the presence of the circadian rhythm of the GPx enzyme.

### 3.6. The Circadian Rhythm of R-GSSG

R-GSSG is an oxidoreductase that catalyzes the reduction of oxidized glutathione (glutathione disulfide, GSSG) using nicotinamide adenine dinucleotide phosphate (NADPH) as a substrate for the production of reduced GSH [56]. GSH is an antioxidant molecule that is a representative redox indicator for the cellular environment of aerobic organisms. It represents the first line of defense against ROS [22] and displays antiapoptotic, anti-inflammatory, and antiproliferative effect [57]. R-GSSG is also an enzyme that protects erythrocytes from hemolysis [58]. In the case of riboflavin deficiency, the activity of R-GSSG decreases, inducing oxidative stress and hemolytic anemia [56].

The results of our own research showed that both in the case of women and men, the average R-GSSG activity is the highest at 2 a.m. As for the average lowest R-GSSG activity, it was recorded at 8 p.m, i.e., 18 h later. Comparing our results with the results of scientists who conducted the study on 60 clinically healthy volunteers and 30 patients with cirrhosis of the liver, it can be noticed that they differ. According to human studies from 2005, R-GSSG peaked between 6 a.m., and 12 a.m., indicating a difference of several hours. The lowest activity of the enzyme was demonstrated at midnight. The maximum and minimum peaks of R-GSSG activity in healthy subjects and those struggling with liver cirrhosis coincided. However, the activity values were lower in the patients [53]. The same scientists repeated the study of the daily R-GSSG activity 10 years later, but on a group of 60 clinically healthy volunteers and 50 patients diagnosed with peptic ulcers of the stomach. The study showed circadian variability in this enzyme’s activity in clinically healthy volunteers and those suffering from gastric ulcer disease. The conclusions drawn from these tests were similar to those of 2005 for patients with liver cirrhosis. The minimum and maximum peaks of the daily R-GSSG activity coincided in the healthy and the sick, with lower activities of this enzyme in patients with gastric ulcer disease, compared to clinically healthy volunteers. Thus, the results of the obtained studies indicate the participation of ROS in tissue damage and, thus, ulceration, resulting from a general decline in defense mechanisms [59]. These tests were also carried out in erythrocytes, excluding differences in the tested material. Still, these scientists should have included information about the method they used for the research in the publication. This allows for the assumption that this method differed most from ours. In addition, these differences may be due to the fact that these scientists adopted different collection times (6 a.m., 12 a.m., 6 p.m., and 12 p.m.), which does not exclude that if we had performed the measurements at the same hours, we would have obtained very similar results because the upward and downward trends were similar depending on the time of day/night. Additionally, in this case, we can observe a peak of activity at night. During sleep, the body removes the ROS accumulated throughout the day. Despite the differences between the results of our research and the results of scientists [53,59], we confirmed the presence of the R-GSSG circadian rhythm after statistical analysis.

### 3.7. The Circadian Rhythm of GST

GST is an enzymatic antioxidant widely distributed in nature. Its task is to conjugate glutathione with xenobiotics, which decrease reactivity, increase solubility, and are more easily removed from the body [60]. Some GSTs can protect lipids from peroxidation, others (e.g., GST A1-1 and A1-2) show activity similar to PHGPx4 in association with the lipid peroxide membrane, and still others (e.g., GST A4-4) may metabolize toxic products terminal lipid peroxidation such as 4-hydroxynonenal (HNE) and A2/J2 isoprostanes [61]. GSTs also have an important role in cellular signaling and redox homeostasis [62,63].

The results of our research showed that both in the case of women and men, the average GST activity is the highest at 2 a.m. Exactly 12 h later, the mean lowest enzyme activity was observed, i.e., at 2 p.m. The first studies on the circadian variability of GST were carried out in Swiss Webster mice. It was then shown that the enzyme’s highest activity was during the dark phase, while the lowest was during the light phase, which would indicate the similarity of the results obtained in our research [55]. Further studies on the hepatic cytosolic fraction of male and female mice showed the highest activity of GST at 1 p.m. [64]. For our research, the most increased GST activity was at 2 a.m., 13 h later. Scientists observed the lowest activity of the enzyme at 9 p.m., so the results do not coincide with our observations. This may be due to another method of testing the enzyme and the fact that the tests were carried out on mouse tissue, not human blood, as in our case. No information has been found in the literature on the activity of GST in the circadian rhythm in humans, so we cannot compare the results. When comparing the results of own research with the results of research carried out on mice [55], it should be remembered that these animals are model organisms and do not always reflect the processes in the human body. Interestingly, the differences in the studies and the statistical analysis carried out did not confirm the occurrence of circadian rhythms in humans.

### 3.8. The Circadian Rhythm of GSH

GSH can occur in mammalian cells in the reduced (GSH) and oxidized (GSSG) forms. GSH is used in the GPx-catalyzed reaction where H_2_O_2_ is reduced, and GSSG is formed GSH is considered a cofactor of antioxidant enzymes and an independent antioxidant [21].

The first tests were performed on male Swiss Webster mice. Scientists have shown that glutathione peaks between 6 a.m. and 10 a.m., which is slightly different from the results obtained in our study [55]. The lowest concentration of glutathione occurred at 6 p.m., which, compared to our results, occurred only 2 h earlier, and this difference may be due to different hours of blood sampling. When comparing the following results of tests performed on mice obtained in 1999 by another team of scientists, slight differences in the highest and the lowest concentration of GSH can be noticed [64]. For these scientists, the lowest concentration occurred at 5 p.m., while in our study, it was 8 p.m. The difference may also be due to the different blood collection times (scientists accepted 1 a.m., 5 a.m., 9 a.m., 1 p.m., 5 p.m., and 9 p.m.). The peak glutathione concentration in our research was at 2 a.m., and the results of the scientists’ research indicate 9 a.m. The results obtained by the scientists who first re-searched humans concerning the GSH enzyme are similar to those obtained in their study [65]. The scientists investigated the effect of melatonin on GSH levels in erythrocytes. Blood samples were collected at 10 a.m. and 10 p.m.—at the beginning of melatonin secretion. They showed a marked increase in melatonin in the circadian variability of GSH in erythrocyte samples late in the evening at 10 p.m. and a decrease at 10 a.m. In our research, the concentration peak was at 2 a.m., and in the morning, it dropped significantly (8 a.m.). This difference may be due to different hours of collection because, in our research, blood was collected every 6 h, and in the scientists’ research every 12 h. Both own and scientists’ research were carried out in erythrocytes, excluding differences in the material tested. The results of these studies [65] and their studies indicate the presence of a daily regulation of GSH concentration.

### 3.9. Relationship between Circadian Rhythms and the Body’s Oxidative Stress—Conclusion

Our research has confirmed the existence of circadian regulation of all examined oxidative stress enzymes, except for GST activity. It was noticed that analyzed enzyme activities (SOD-1, GPx, CAT, and R-GSSG) and melatonin and GSH concentrations peaked at 2:00 am. The highest activity of these enzymes and the highest concentration of GSH at night are most likely related to the intensification of processes leading to the removal of free radicals during sleep, which in turn leads to the inhibition of oxidative stress in the organism. Our observations shed more light on understanding oxidative processes against the background of circadian rhythms in the human body. In the era of dynamic development of regenerative medicine, these are issues of considerable importance and may contribute to a more efficient diagnosis of patients and the selection of more effective treatment in diseases that may result from the overproduction of free radicals, leading to oxidative damage of cellular components. We also wonder whether a strictly defined period during the day may affect the medications’ effectiveness. Although we know these considerations are purely speculative, they give us reasons to expand our research soon.

## 4. Materials and Methods

### 4.1. Study Group

The research was carried out in a group of 66 healthy volunteers aged 20 to 50 years. This group was divided according to gender (33 women with a mean age of 31 ± 7 years and 33 men with a mean age of 34 ± 10 years). Before taking them, they completed a questionnaire on their health, lifestyle, physical activity, medications/supplements, and sleep duration. The following exclusion criteria were used: pregnancy, taking medications used in the treatment of chronic diseases, contraceptives, dietary supplements, especially antioxidants, i.e., vitamin C or vitamin E, the use of hormone replacement therapy, as well as antibiotics and painkillers in a period of fewer than 2 weeks before the tests, and a multi-shift work system. A particular room with a sleeping bed was provided to maintain the correct sleep rhythm and wakefulness during blood donation.

### 4.2. Study Material

For the tests, peripheral blood was collected from the antecubital vein four times at 2 a.m., 8 a.m., 2 p.m., and 8 p.m. To obtain the biological material used for the research (erythrocytes and serum), blood was collected by educated medical personnel of the Independent Public Clinical Hospital No. 1 in Szczecin on K_2_EDTA and for a clot, respectively.

Peripheral blood counts (ABX Micros 60 Hematology Analyzer, Kyoto, Japan) were performed in each patient’s samples collected on K_2_EDTA. Then, the blood collected on K_2_EDTA was centrifuged, and the obtained erythrocyte fraction was washed three times with a PBS solution. The obtained erythrocytes were frozen at −80 °C until the tests were performed. To confirm the proper health of the volunteers, we performed additional tests in the centrifuged blood serum—basic biochemical parameters and the concentration of minerals. All these determinations were performed based on BioMaxima reagent kits by the procedures suggested by this manufacturer (BioMaxima, Lublin, Poland). Deviation from the established reference values for individual parameters was an additional criterion excluded from the study.

### 4.3. Determination of Melatonin Concentration

To confirm the maintenance of the correct rhythm of sleep and wakefulness, the concentration of melatonin was measured. These tests were performed in blood serum at all sampling time points. For this purpose, the competitive reagent test Elisa-Human Melatonin ELISA Kit, Cloud-Clone Corp, Houston, TX, USA was used. The determinations were made according to the manufacturer’s instructions attached to the kit. Absorbance readings were taken with an EnVision microplate reader (Perkin Elmer, Waltham, MA, USA) at 450 nm.

### 4.4. Determination of Hemoglobin Concentration by the Drabkin Method in Erythrocyte Hemolysates

All results obtained for the activity of enzymes were calculated per 1 g of protein, in this case, hemoglobin. Therefore, hemoglobin was determined in erythrocytes using the standard and well-known Drabkin method for this purpose. A UV/VIS Lambda 40 spectrometer was used to measure the extinction of individual samples.

### 4.5. Determination of Superoxide Dismutase Activity (CuZn-SOD—SOD-1) in Erythrocytes

To determine the SOD-1 activity in erythrocytes, a UV/VIS Lambda 40 spectrometer was used. In the samples of erythrocytes brought to room temperature, hemoglobin was determined using the Drabkin method based on its concentration. Erythrocyte hemolysate was prepared at a concentration of 5 g/dL. SOD-1 was extracted from erythrocytes by adding 0.25 mL of hemolysate 5 g/dL and 0.2 mL of a mixture of chloroform and ethanol in the ratio of 3:5 and 0.3 mL of distilled H_2_O followed by 5 min centrifugation (4 °C, 6000 rpm). The obtained extract was used for SOD-1 extinction measurements in the tested erythrocytes after adding 1.475 mL of 0.05 M carbonate buffer at pH 10.2 and 0.025 mL of adrenaline solution dissolved in 0.1 M HCl against the blank (1.475 mL of 0.05 M buffer carbonate at pH 10.2 and 0.025 mL of extract). The extinction of the test sample was measured for 5 min at a wavelength of 320 nm and preceded by a 3 min incubation at 30 °C. Additionally, an extinction measurement was a performer for the control sample (1.45 mL of 0.05 M carbonate buffer at pH 10.2 and 0.05 mL of adrenaline solution in 0.1 M HCl) against the blank (1.5 mL of 0.05 M carbonate buffer at pH 10.2) [66,67].

### 4.6. Determination of Catalase Activity (CAT) in Erythrocytes

A UV/VIS Lambda 40 spectrometer was used to determine CAT activity in erythrocytes. In samples of erythrocytes brought to room temperature, hemoglobin was determined using the Drabkin method based on its concentration. Erythrocyte hemolysate was prepared at a concentration of 5 g/dL. This hemolysate in 10 µL was added to 5 mL of 50 mM phosphate buffer pH 7.0. Then, 1 mL of this hemolysate was used for CAT extinction measurements in the test erythrocytes after adding 0.5 mL of 30 nM H_2_O_2_ solution to the blank (1 mL of hemolysate + 0.5 mL of 50 mM phosphate buffer at pH 7.0). Extinction measurements of the test sample were performed for 30 s in quartz cuvettes at a wavelength of 240 nm [66,67].

### 4.7. Determination of Glutathione Peroxidase (GPx) Activity in Erythrocytes

To determine GPx activity in erythrocytes, a UV/VIS Lambda spectrophotometer was used. In the samples of erythrocytes brought to room temperature, hemoglobin was determined using the Drabkin method based on its concentration. Erythrocyte hemolysate was prepared at a concentration of 3 g/dL. This hemolysate, in an amount of 0.5 mL, was added to 0.25 mL of the transforming reagent (4.5 mM KCN; 0.45 Mm K_3_[Fe(CN)*_6_*] and incubated at room temperature for 5 min. Then, for the test samples and the blank sample, a reaction mixture was prepared to consist of 0.55 mL of 50 mM phosphate buffer at pH 7.0, 0.05 mL of glutathione reductase solution in phosphate buffer with the activity of 6IU, 0.05 m: of 10 mM reduced glutathione (GSH), 0.05 mL of 2.5 mM NADPH + H + in 0.1% NaHCO_3_ solution and 0.25 mL of 3 g/dL hemolysate with transformation reagent. The measurement of the extinction of the test sample was carried out for 5 min at 340 nm in the presence of 0.25 mL of bi-distilled H_2_O as a blank. A 10 min incubation preceded it at 37 °C and added 0.05 mL of 12 mM tert-butyl hydroxide (T-BOOH) was. The rate of the non-enzymatic reaction was measured by measuring in a system containing water instead of hemolysate [66,67].

### 4.8. Determination of Glutathione Reductase (R-GSSG) Activity in Erythrocytes

To determine the activity of R-GSSG in erythrocytes, a UV/VIS Lambda spectrophotometer was used. A hemolysate was prepared from the erythrocytes brought to room temperature (100 µL of erythrocytes + 2 mL of bi-distilled H_2_O). To 0.025 mL of hemolysate, 1 mL of RI (0.05 mL of 0.006 M NADPH + H + solution in 0.01 M NaOH, 0.95 mL of triethanolamine–EDTA buffer at pH 7.5) was added, and incubated for 5 min at 30 °C. After the incubation time had elapsed, 0.1 mL of RII reagent (0.19 mL triethanolamine–EDTA buffer pH 7.5, 0.01 mL of 0.045 M GSSG oxidized glutathione solution) was added, and extinction was measured at 340 nm for 5 min in a temperature of 30 °C [66,67].

### 4.9. Determination of Glutathione Transferase (GST) Activity in Erythrocytes

The UV/VIS Lambda 40 spectrophotometer was used to determine the GST activity in erythrocytes. In the samples of erythrocytes brought to room temperature, hemoglobin was determined using the Drabkin method and based on its concentration. Erythrocyte hemolysate was prepared at a concentration of 5 g/dL. Then, a reaction mixture was prepared to consist of 0.85 mL for the test sample and 0.9 mL for the controls sample 0.1 M phosphate buffer pH 6.5, 0.05 mL 20 mM GSH solution, 0.05 mL 20 mM CDNB solution, and 0.05 mL of hemolysate 5 g/dL in the case of the test sample. The measurement of the extinction of the test sample was carried out for 5 min at 340 nm in the presence of bi-distilled H_2_O as 1 mL of a blank. Additionally, an extinction measurement was performed for the control sample against bi-distilled H_2_O as a blank [66,67].

### 4.10. Determination of Glutathione (GSH) Concentration in Erythrocytes

To determine the concentration of GSH in erythrocytes, a UV/VIS Lambda 40 spectrophotometer was used. In the samples of erythrocytes brought to room temperature, hemoglobin was determined using the Drabkin method based on its concentration. Erythrocyte hemolysate was prepared at a concentration of 5 g/dL. Then, a reaction mixture was prepared to consist of 0.45 mL of bi-distilled H_2_O, 0.75 mL of precipitating solution (1.67 g of glacial metaphosphoric acid, 0.2 g of disodium/dipotassium EDTA, 30 g of NaCl in 100 mL of bi-distilled H_2_O and 0.1 mL of hemolysate 5 g/dL. This mixture was incubated for 5 min at 4 °C and then centrifuged (4 °C, 5500 rpm, 10 min). Then, 1 mL of supernatant and 0.5 mL of DTNB solution were added to 4 mL of phosphate buffer at pH 7.9 and incubated for 15 min at 4 °C. The extinction was determined at a wavelength of 412 nm, at the temperature of 25 °C against the blank (0.6 mL of precipitating solution, 0.4 mL of bi-distilled H_2_O, 4 mL of phosphate buffer at pH 7.9, 0.5 mL of DTNB solution) [66,67].

### 4.11. Statistical Analysis

The results obtained during the tests were statistically analyzed using the Statistica PL 13 Trial (StatSoft) software, Cracow, Poland. The normality of distributions was checked using the Schapiro–Wilk test. In some cases, the variables differed from the normal distribution of parameters. Each tested parameter was characterized by arithmetic mean, standard deviation, group size, median, lower and upper quartiles, and the range between quartiles. The MANOVA test assessed the differences between the studied parameters depending on gender. Friedman’s ANOVA test, as well as the Wilcoxon pairwise test, was used to assess the differences between the studied parameters. To confirm the presence of the circadian rhythm, the analysis of the Kendall concordance coefficient was performed. The Fisher and Chi-square tests were used for the study of qualitative data. The strength of the correlation between melatonin and individual parameters (SOD-1, CAT, GPx, R-GSSG, GST, GSH) was measured thanks to the Spearman rank coefficient. The *p*-value < 0.05 was considered statistically significant.

## Figures and Tables

**Figure 1 ijms-23-14275-f001:**
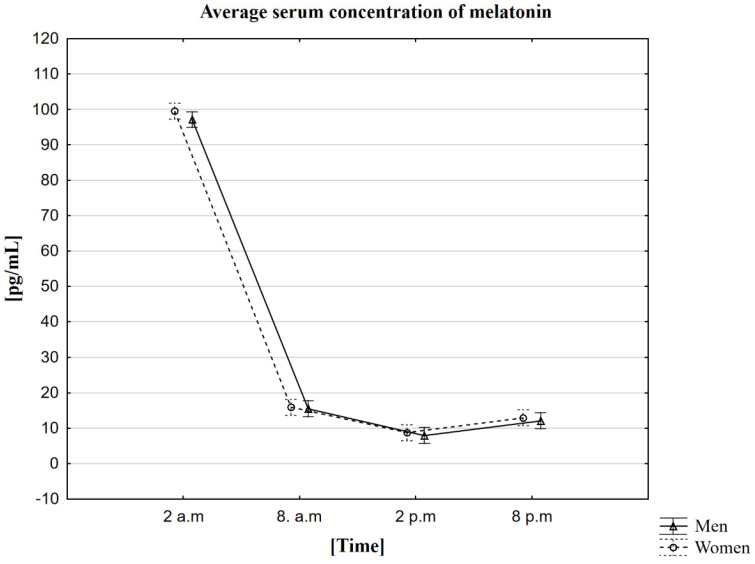
The average serum concentration of melatonin in men and women at different sampling time points. Data presented as mean ±95% confidence interval. MANOVA-type analysis (*p* = 0.82172).

**Figure 2 ijms-23-14275-f002:**
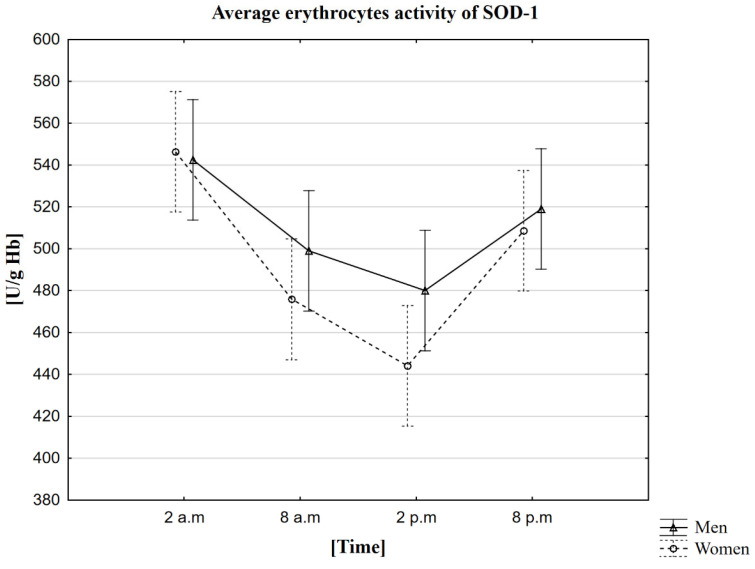
Average erythrocytes activity of SOD-1 in men and women at different sampling time points. Data presented as mean ± 95% confidence interval. MANOVA-type analysis (*p* = 0.56135).

**Figure 3 ijms-23-14275-f003:**
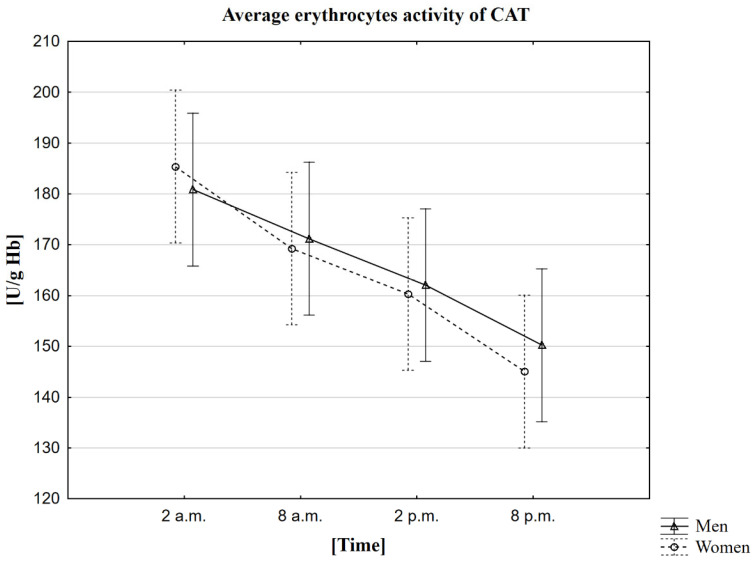
Average erythrocytes activity of CAT in men and women at different sampling time points. Data presented as mean ± 95% confidence interval. MANOVA-type analysis (*p* = 0.93485).

**Figure 4 ijms-23-14275-f004:**
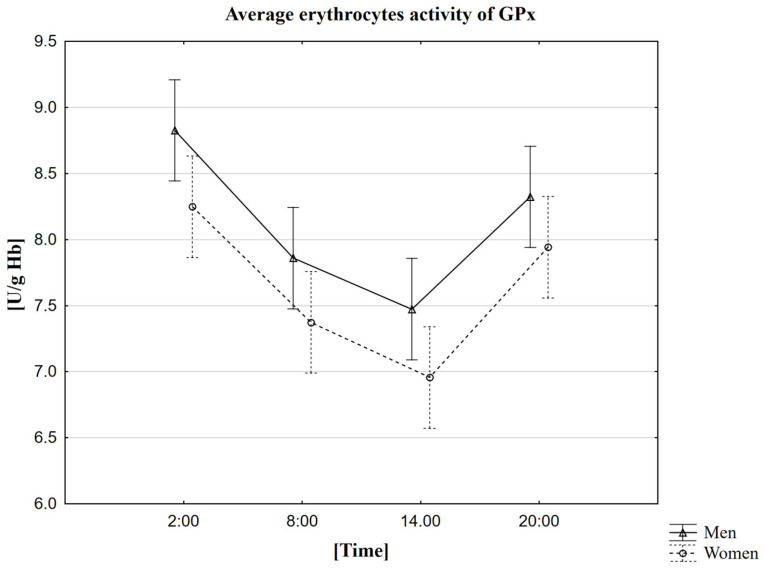
Average erythrocytes activity of GPx in men and women at different sampling time points. Data presented as mean ±95% confidence interval. MANOVA-type analysis (*p* = 0.96529).

**Figure 5 ijms-23-14275-f005:**
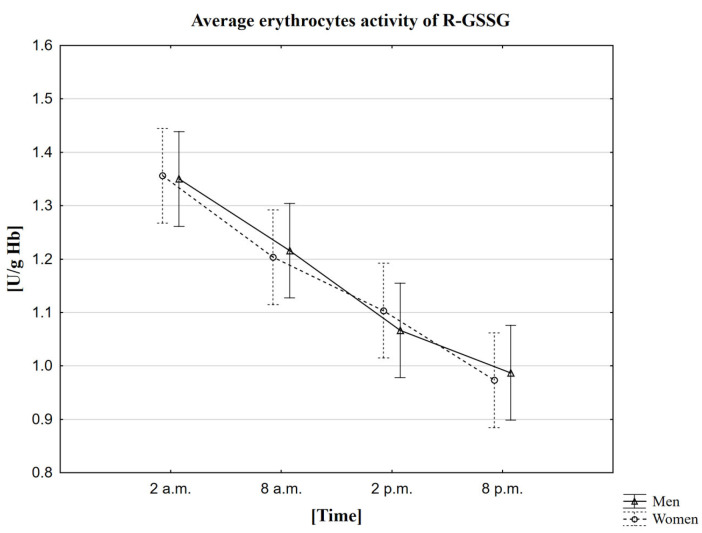
Average erythrocytes activity of R-GSSG in men and women at different sampling time points. Data presented as mean ±95% confidence interval. MANOVA-type analysis (*p* = 0.93669).

**Figure 6 ijms-23-14275-f006:**
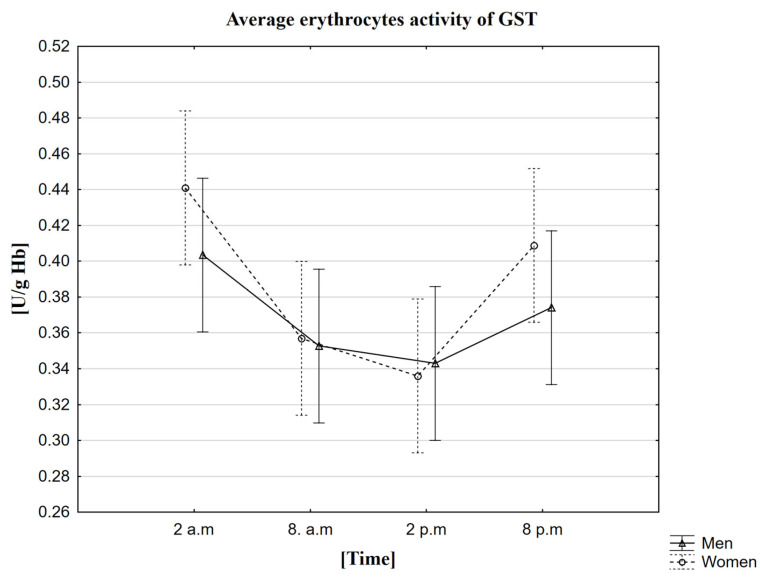
Average erythrocytes activity of GST in men and women at different sampling time points. Data presented as mean ± 95% confidence interval. MANOVA-type analysis (*p* = 0.67233).

**Figure 7 ijms-23-14275-f007:**
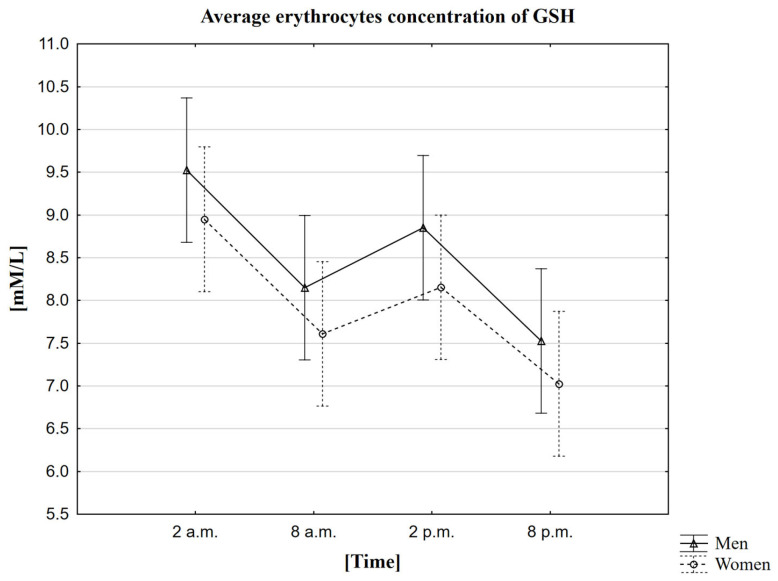
Average erythrocytes concentration of GSH in men and women at different sampling time points. Data presented as mean ± 95% confidence interval. MANOVA-type analysis (*p* = 0.99631).

**Table 1 ijms-23-14275-t001:** Mean concentration of blood parameters obtained among volunteers divided by gender, *p* < 0.05.

Parameters	Women (n = 33)	Men (n = 33)	*p*-Value
	Value (Mean ± SD) [NR]	Value (Mean ± SD) [NR]	
RBC [10^12^/L]	4.5 ± 0.4 [3.8–5.2]	5.3 ± 0.5 [4.0–5.9]	**<0.0001**
Hb [mM/L]	7.9 ± 0.6 [7.4–9.9]	9.2 ± 0.6 [8.7–11.2]	**<0.0001**
Hct [%]	40 ± 3 [33–44]	46 ± 3 [41–50]	**<0.0001**
WBC [10^9^/L]	6.0 ± 1,4 [4.5–11.0]	5.8 ± 1,1 [4.5–11.0]	0.9929
PLT [10^3^/µL]	252 ± 41 [150–450]	253 ± 40 [150–450]	0.9715
GLC [mg/dL]	90 ± 8 [70–99]	92 ± 7 [70–99]	0.1311
IP [mM/L]	1.5 ± 0.2 [1.12–1.45]	1.5 ± 0.1 [1.12–1.45]	0.9359
tMG [mM/L]	0.92 ± 0.03 [0.85–1.10]	0.93 ± 0.04 [0.85–1.10]	0.6487
TCAL [mM/L]	2.3 ± 0.2 [2.2–2.6]	2.4 ± 0.1 [2.2–2.6]	0.7141
TC [mg/dL]	179 ± 23 [125–200]	187 ± 22 [125–200]	0.1555
TG [mg/dL]	92 ± 19 [<150]	111 ± 35 [<150]	**0.0207**
LDL-C [mg/dL]	89 ± 23 [50–100]	92 ± 25 [50–100]	0.7141
HDL-C [mg/dL]	70 ± 11 [50–90]	62 ± 14 [45–70]	0.3669
TP [g/dL]	6.5 ± 0.4 [6–8]	6.6 ± 0.5 [6–8]	0.2491
Alb [g/dL]	3.9 ± 0.3 [3.5–5.0]	4.0 ± 0.3 [3.5–5.0]	0.0525
CRE [mg/dL]	0.9 ± 0.2 [0.6–1.1]	1.0 ± 0.2 [0.7–1.3]	**0.0012**
UA [mg/dL]	4.5 ± 0.7 [2.7–7.3]	5.3 ± 1.0 [4.0–8.5]	**0.0005**

Abbreviations: SD—standard deviation, NR—normal range, RBC—red blood cell count, Hb—hemoglobin, Hct—hematocrit, WBC—white blood cells count, GLC—glucose, IP—inorganic phosphorus, tMg—total magnesium, TCAL—total calcium, TC—total cholesterol, TG—triglycerides, LDL-C—low-density lipoprotein-cholesterol, HDL-C—high-density lipoprotein-cholesterol, TP—total protein, Alb—albumin, CRE—creatinine, UA—uric acid.

**Table 2 ijms-23-14275-t002:** Parameters describing melatonin concentration in women and men, including the time of blood collection [pg/mL].

Gender	Women (n = 33)				Men (n = 33)			
Time of Blood Collection	2 a.m.	8 a.m.	2 p.m.	8 p.m.	2 a.m.	8 a.m.	2 p.m.	8 p.m.
Mean	99.5	15.9	8.7	12.9	97.1	15.5	8.0	12.1
Standard deviation	±11.2	±3.8	±2.5	±4.7	±11.7	±4.4	±2.6	±3.0
Median	98.4	15.3	9.1	12.2	98.1	14.5	8.1	12.2
Minimum	76.2	8.3	3.6	4.7	72.1	8.2	3.9	5.7
Maximum	135.8	22.8	13.4	27.1	118.9	23.8	15.0	19.0
Upper quartile	105.3	18.2	10.3	15.1	104.6	18.8	10.1	13.8
Lower quartile	94.6	13.9	6.9	9.4	89.4	12.0	5.8	10.3
Interquartile range	10.7	4.3	3.4	5.7	15.2	6.8	4.3	3.5

**Table 3 ijms-23-14275-t003:** Parameters describing the activity of superoxide dismutase (SOD-1) in women and men including the time of blood collection [U/g Hb].

Gender	Women (n = 33)				Men (n = 33)			
Time of Blood Collection	2 a.m.	8 a.m.	2 p.m.	8 p.m.	2 a.m.	8 a.m.	2 p.m.	8 p.m.
Mean	546	476	444	509	542	499	480	519
Standard deviation	±85	±74	±76	±78	±91	±87	±85	±93
Median	544	473	427	501	536	501	479	522
Minimum	398	371	315	382	388	364	356	370
Maximum	753	737	715	763	758	724	713	740
Upper quartile	595	529	492	558	580	532	512	551
Lower quartile	514	418	399	453	477	440	432	447
Interquartile range	81	111	94	105	103	92	80	104

**Table 4 ijms-23-14275-t004:** Parameters describing the activity of catalase (CAT) in women and men including the time of blood collection (U/g Hb).

Gender	Women (n = 33)				Men (n = 33)			
Time of Blood Collection	2 a.m.	8 a.m.	2 p.m.	8 p.m.	2 a.m.	8 a.m.	2 p.m.	8 p.m.
Mean	185	169	160	145	181	171	162	150
Standard deviation	±49	±43	±43	±40	±45	±44	±43	±43
Median	173	159	149	136	176	160	150	142
Minimum	122	108	92	91	99	95	86	83
Maximum	285	275	271	231	281	268	267	248
Upper quartile	210	201	186	165	213	201	192	179
Lower quartile	155	140	131	117	152	142	137	123
Interquartile range	55	61	55	48	61	59	55	56

**Table 5 ijms-23-14275-t005:** Parameters describing glutathione peroxidase (GPx) activity in women and men, including the time of blood collection (U/g Hb).

Gender	Women (n = 33)				Men (n = 33)			
Time of Blood Collection	2 a.m.	8 a.m.	2 p.m.	8 p.m.	2 a.m.	8 a.m.	2 p.m.	8 p.m.
Mean	8.2	7.4	7.0	7.9	8.8	7.9	7.5	8.3
Standard deviation	±1.4	±1.2	±1.1	±1.3	±1.0	±1.0	±0.9	±1.0
Median	8.6	7.6	7.3	8.2	8.9	7.7	7.5	8.3
Minimum	5.5	5.3	5.1	5.6	7.2	6.3	5.5	6.8
Maximum	10.6	9.5	9.1	10.0	10.8	10.0	9.4	10.1
Upper quartile	9.2	8.2	7.6	9.0	10.0	8.4	8.1	8.2
Lower quartile	7.3	6.3	6.0	7.0	8.1	7.1	7.0	7.6
Interquartile range	1.9	1.9	1.6	2.0	1.9	1.3	1.0	1.2

**Table 6 ijms-23-14275-t006:** Parameters describing the activity of glutathione reductase (R-GSSG) in women and men, including the time of blood collection (U/g Hb).

Gender	Women (n = 33)				Men (n = 33)			
Time of Blood Collection	2 a.m.	8 a.m.	2 p.m.	8 p.m.	2 a.m.	8 a.m.	2 p.m.	8 p.m.
Mean	1.4	1.2	1.1	1.0	1.4	1.2	1.1	1.0
Standard deviation	±0.3	±0.3	±0.3	±0.2	±0.3	±0.3	±0.2	±0.2
Median	1.3	1.1	1.0	0.9	1.3	1.2	1.1	0.9
Minimum	0.8	0.8	0.6	0.6	0.8	0.7	0.6	0.6
Maximum	2.0	1.8	1.7	1.6	1.9	2.0	1.5	1.6
Upper quartile	1.4	1.3	1.2	1.1	1.5	1.4	1.2	1.1
Lower quartile	1.2	1.0	0.9	0.9	1.2	1.1	1.0	0.8
Interquartile range	0.2	0.3	0.3	0.2	0.3	0.3	0.2	0.3

**Table 7 ijms-23-14275-t007:** Parameters describing glutathione transferase (GST) activity in women and men, including the time of blood collection (U/g Hb).

Gender	Women (n = 33)				Men (n = 33)			
Time of Blood Collection	2 a.m.	8 a.m.	2 p.m.	8 p.m.	2 a.m.	8 a.m.	2 p.m.	8 p.m.
Mean	0.44	0.36	0.34	0.41	0.40	0.35	0.34	0.37
Standard deviation	±0.14	±0.12	±0.13	±0.17	±0.09	±0.10	±0.12	±0.11
Median	0.44	0.35	0.32	0.40	0.39	0.34	0.34	0.32
Minimum	0.19	0.15	0.11	0.16	0.28	0.19	0.11	0.25
Maximum	0.78	0.66	0.59	0.99	0.66	0.59	0.59	0.69
Upper quartile	0.50	0.42	0.45	0.44	0.43	0.42	0.41	0.43
Lower quartile	0.34	0.30	0.25	0.30	0.34	0.27	0.27	0.30
Interquartile range	0.16	0.12	0.20	0.14	0.09	0.15	0.15	0.14

**Table 8 ijms-23-14275-t008:** Parameters describing concentration of glutathione (GSH) in women and men, including the time of blood collection (mM/L).

Gender	Women (n = 33)				Men (n = 33)			
Time of Blood Collection	2 a.m.	8 a.m.	2 p.m.	8 p.m.	2 a.m.	8 a.m.	2 p.m.	8 p.m.
Mean	9.0	7.6	8.2	7.0	9.5	8.2	8.9	7.5
Standard deviation	±2.7	±2.5	±2.7	±2.6	±2.4	±2.2	±2.4	±2.1
Median	7.9	6.7	7.2	6.0	8.9	7.4	7.8	6.5
Minimum	5.5	5.2	5.4	5.0	6.1	5.1	5.3	5.1
Maximum	16.1	14.5	15.6	14.2	13.9	12.4	13.5	12.3
Upper quartile	9.7	7.9	8.4	7.2	11.1	10.2	10.8	9.7
Lower quartile	7.1	6.1	6.4	5.3	7.8	6.5	7.2	6.1
Interquartile range	2.6	1.8	2.0	1.9	3.3	3.7	3.6	3.6

## Data Availability

The data presented in this study are available on request from the corresponding author. The data are not publicly available due to institutional privacy restrictions.

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
