# Peer review of "The Influence of Circadian Rhythm on the Activity of Oxidative Stress Enzymes"

_ijms, 2022, doi:10.3390/ijms232214275_

Round 1

Reviewer 1 Report

In this manuscript, the authors indicate the influence of circadian rhythm on the activity of oxidative stress enzymes. The manuscript can be accepted after addressing the below mentioned corrections.

1.     P2-L56: After sentences ‘Over millions of years of evolution, organisms have developed various protective systems…’’ It should be given information. For this purpose the authors can look at the following articles for introduction section: Journal of pharmacy and pharmacology 71 (10), 1576-1583 and Drug development research 81 (5), 628-636.

2.     P2-L64: After sentences ‘In turn, the presence and activity of cellular antioxidant enzymes are the basic defense mechanism against…’’ It should be given information. For this purpose the authors can look at the following articles for introduction section: Environmental toxicology and pharmacology 70, 103195 and Archives of physiology and biochemistry 125 (3), 235-243.

3.     P3-L113 After sentences ‘R-GSSG is an oxidoreductase that catalyzes the …’’ It should be given information. For this purpose the authors can look at the following articles for introduction section: Protein and peptide letters 26 (5), 364-370 and Biotechnology and Applied Biochemistry 69 (1), 281-288.

4.     P3-L137, After sentences ‘GST is an enzyme whose action is based on the coupling …’’ It should be given information. For this purpose the authors can look at the following articles for introduction section: Journal of biochemical and molecular toxicology 32 (9), e22196, ChemistrySelect 6 (43), 11915-11924 and ChemistrySelect 6 (40), 11137-11143, 

5.     P3-L142, After sentences ‘’ Reduced GSH plays a key role…’’ It should be given information. For this purpose the authors can look at the following articles for introduction section: Journal of biochemical and molecular toxicology 32 (5), e22047 and Journal of biochemical and molecular toxicology 31 (11), e21967

6.     What is the reason for the high standard deviation in the figures?

7.     Grammaticals errors found in the manuscript. It should be corrected.

Reviewer 2 Report

The article submitted by Marta Budkowska and co-workers entitled “The influence of circadian rhythm on the activity of oxidative stress enzymes” addresses the role of superoxide dismutase (SOD), glutathione peroxidase (GPx), catalase (CAT), glutathione transferase (GST), and glutathione reductase (R-GSSG) in enzymatic antioxidant defense associated with circadian rhythm. This article represents a well-structured study and addresses an important issue of biomedical research. The authors have high expertise in the field, and the reported data are of sufficient novelty. However, several points should be addresed before publication:

Major points:

The Introduction section is too long and looks like a review of the previously obtained or known data. Here, the authors should modify (shorten) the text specifying mainly the goals of the study and the methods how they could be achieved. Other information provided in this section could be translocated to the results and discussion section.

Minor points:

1) P. 2, line 73: in the sentence “…to neutralize O2°..” is little confusing. If it is a superoxide, it should be present in its anionic form O2-

2) P. 2, line 77: there is a misprint describing the function of catalase. This enzyme decomposes H2O2.

3) The tittle of table 4 does not correspond to its content (please see text in lines 218-222).

4) P. 16, line 427: in the sentence “…because its main function in all aerobic organisms is to O2 - into H2O2..” the word “convert” is missing. Here again, superoxide should be present in its anionic form O2-.

5) P. 17, line 462: lower indexes should be used in the formula of H2O2

6) P. 17, line 479: the sentence “…The results of my own research..” should be changed to the sentence “…The results of our own research..”

7) P. 20, line 625: the sentence “The activity of enzymes was converted into 1 mg of protein.” is awkward and should be rephrased.

8) P. 21, line 661: lower indexes should be used in the formula of K3Fe(CN) 6).

Reviewer 3 Report

The work is devoted to the study of the influence of circadian rhythms on the activity of oxidative stress enzymes. It is done on a good methodological level. The data obtained are extremely important for understanding oxidative processes against the background of circadian rhythms in the human body. The obtained data also comprehended in terms of comparison with numerous studies on the study of circadian rhythms for the content of melatonin and glutathione, as well as the activity of superoxide dismutase, catalase, glutathione peroxidase, glutathione reductase, glutathione transferase. In some cases, the data obtained differ significantly in terms of the time of maximum activity of enzymes and the content of the studied compounds.

In my opinion, the work lacks a Conclusion section, or a subsection in the Discussion, within which the obtained results would be comprehended in the complex of the relationship between circadian rhythms and the body's oxidative stress. The most important result, indicating that in the study for men and women for both the concentration of melatonin and the activity of antioxidant defense enzymes and the concentration of glutathione, the maximum peak values occur at 2 a.m., is noted in the abstract, but not discussed in the article.

However, this comment, as well as the comments below, do not reduce the value of the work. After their corrections, the work can be published in IJMS.

Comments

The size of the abstract significantly exceeds the size of “about 200 words maximum” established by the IJMS rules. Shorten the abstract to the prescribed length of 200 words maximum.

Lines 178, 368

Reword the title of paragraphs 2.1 and 3.1. For example, on “morphological, biochemical, and blood minerals parameters”. In its current form, it is not clear from the name which minerals are in question.

Lines 590-592

“The research was carried out in a group of 66 healthy volunteers aged 20 to 50 years. 590 This group was divided according to gender (33 women with a mean age of 31 ± 7 years and 33 men with a mean age of 34 ± 10 years).”

How did you get the scatter of the 20-50 year group if the age for women was 31 ± 7 years, and for men 34 ± 10 years? With such values, the spread in age of all volunteers should have been 24-44 years.

Table 2 and Figure 1

The data in Table 2 and Fig. 1, on the one hand, should duplicate the information on the content of melatonin, on the other hand, they contradict each other. From figure 1 it follows that the highest content of melatonin is at 8 pm, and from table 2 it follows that the highest content of melatonin is at 2 am. Perhaps the timeline is shifted in Figure 1. Based on the values given in Table 2, for the data in Figure 1 the signatures should be: 8 a.m.; 2 p.m.; 8 p.m.; 2 a.m. Reformulate the data presented, correct the figure captions, or rearrange the graph for Figure 1.

Table 3 and Figure 2

There is also a discrepancy between the initial data and after their processing as a result of MANOVA-Type Analysis. Perhaps the timeline has shifted in Figure 2.

Table 7 and Figure 6

Data mismatch. Perhaps the timeline is shifted in Figure 6. Table values for 2 a.m. in the figure are for 2 p.m.

Round 2

Reviewer 1 Report

The manuscript can be accepted this form.

Author Response

We would like to inform You that we have received the following Academic Editor Notes:

,,Reviewer recommended that the authors add citations of 10 reviewer's own papers which are only indirectly related to the topic of the reviewed article. The authors should not have aggreed to this non-ethical request. I therefore recommend acceptance after the authors delete the newly added references 16, 17, 21, 35, 36, 64-68”

Therefore, in the face of this recommends, we have decided to delete the suggested references.

We hope that this revised version of our manuscript despite the lack of these references is still clear, concise and scientifically strengthened.